# Tackling Gender-Related Violence: How Can Theory Inform International Professional Education Projects?

gigi guizzo [1,*,†] and Pam Alldred [2,*,†]

1    RINOVA, Polo Digital, Avenida de Sor Teresa Prat, 29003 Malaga, Spain
2    Department of Social Work, Care and Community, Nottingham Trent University, 50 Shakespeare Street, Nottingham NG1 4BU, UK
*    Correspondence: gigig@rinova.es (g.g.); pam.alldred@ntu.ac.uk (P.A.)
†    These authors contributed equally to this work.

**Abstract:** Is it helpful to share feminist theory with youth practitioners and is there room for it on short training courses such as in EU Action Projects? Can theoretical work on intersectionality, and the concept of gender-related violence (GRV) which grew from it, be shared in training interventions with professionals who work with children and young people? This article is based on the findings of the EU co-funded GAP Work Project that sought to improve GRV intervention and referral through training for practitioners in everyday (rather than specialist) contact with children or young people in four countries. Summarising how the project worked, and how theory informed it, including a brief account of how the concept of GRV worked in practice, guides the selection of material from the wider Project Final Report and offers a reflection on how educators used theory in the training, sometimes explicitly in the sessions. It therefore contributes our experiences to discussions about the design and implementation of education and training about violence and abuse. It concludes by sharing resources for designing and implementing training on sexual harassment, violence and hate crime, including from other recent projects that offer resources for incorporating an intersectional perspective when developing local government plans, programmes, and projects.

**Keywords:** gender-related violence; gender-based violence; professional education and training; intersectionality





## 1. Introduction

This article is based on the final report of the GAP Work Project which aimed to improve intervention and referral on gender-related violence (GRV) through improving the knowledge and understanding of youth practitioners by designing and piloting training. The project was funded by the European Union's DAPHNE-III Programme and led by Brunel University London (UK) between 2013 and 2015, with Dr Pam Alldred as the coordinator.[1] The project was implemented seven years ago, but its experiences remain relevant to many projects that seek to raise awareness of violence, to promote analysis of inequality, to offer feminist interventions, and to do so via training or education. While the environment sometimes seems more hostile at a political level now and social media allows misogynistic influencers, many practitioners and professional bodies continue to do sound work and to seek further interventions to promote equality and to reduce or respond to violence. The issue of using training methodologies when an educational intervention is really what is sought is considered more fully in Cullen and Whelan (2021) on the basis of this project, and in Jones et al. (2021) in relation to a subsequent project.

At the outset of the project, we started to review what was known about the value of relevant training methodologies, but found little published material and although the 'grey literature' of campaigning or charitable organisations was encouraging, it tended to lack detail on training methods. Since the collaborators were all educators (based in

universities) or trainers (in the voluntary/campaign sector), we pooled our expertise and co-designed, with local trainers in each location (four Partners in different countries), four novel training programmes on challenging gender-related violence.

Here we share what we learned about training methodologies and how feminist theory informed the programmes. Our hope is that these understandings can inform how others provide professionals with training and education on gendered violence.

We will first describe the project itself, and then draw out what we feel we learned about using theory in training regarding feminist perspectives and violence awareness, intersectionality and recognising difference, and interactive pedagogies and reflexive methods. In each case, we will explore the various ways the different teams used and shared feminist theory with practitioners, including the challenges this produced.

## 2. The Project and Partners

The GAP Work Project tackled gender-related violence (GRV) in children and young people's lives through improving professional understanding of it and involved 20 organizations from six countries in Europe. The team comprised 11 partner organisations (universities) and 9 associate partners (NGOs and training organisations) across the six countries, and the main actions took place in Ireland, Italy, Spain, and the UK where, in each country, a university lead the activities, overseen by a local action coordinator (coordinator). (In EU co-funded projects it is common to have a distinction between fully funded partners and some with only indirect funding, called associate partners. The four countries above were full partners, but Hungary and Serbia were only associate partners so had expenses for translation (but not for research or training) and adapted programmes that had undergone full piloting elsewhere.) Training materials were developed for practitioners who have everyday contact with children and young people, collectively referred to as 'youth practitioners' in the project. The training actions aimed to improve both the identification of, response to, and prevention of GRV. This aim was to be met by educational interventions and training programmes for practitioners to (i) increase their knowledge and understanding of how best to support children and young people affected by violence, including knowing when and how to refer them to appropriate support services, and (ii) help them to identify and challenge sexist, sexualising, controlling, homophobic, or transphobic language and behaviour in their everyday behaviour and cultural norms.

Professionals outside of legal and welfare services typically receive little training on GRV but given their contact with large numbers of children and young people they urgently need to be better informed. GAP Work Partners surveyed existing training for professionals on GRV and what was known of its effectiveness and then developed innovative training materials for youth practitioners, particularly teachers, youth workers, and healthcare workers. Targeting this general group of practitioners and encouraging them, in turn, to share their learning with their colleagues in a learning 'cascade' was in line with the EU's DAPHNE-III programme priority of delivering training to professionals in contact with victims of violence. GAP Work designed and piloted educational resources for practitioners and trained them to skills-share with their colleagues (via the cascade) in Ireland, Italy, Spain, and the UK, and funded the translation of these materials into two other languages (Serbian and Hungarian) which were checked by local experts of gender and homophobic violence, respectively. Thus, there were four project 'partners' (directly funded) and two 'associate partners' who did not receive funding directly. The full project report and its legacy in terms of training materials can be found here: https://usvreact.eu/gap-work-project/ (accessed on 1 November 2022) but sadly the full project website was taken down when the PI left the university.

The GAP Work initiative aimed to bridge gaps in practice related to gender-related violence (GRV), in particular, the gap between services for *adults* and for *children*, and the gap between *responses* to GRV that has been identified, and *prevention* through educational work. The partners adopted a wide definition of GRV to address sexist, sexualizing, or norm-driven bullying and harassment in children and young people's lives. The initiative

intended to place a critique of gender norms and normativity at the centre to tackle all forms of norm-related violence, but was particularly focused on violence against women and girls, homophobic, lesbophobic, and transphobic violence (see Alldred 2023 for discussion of the origins of the project and definitions, and Biglia's discussions in English and Spanish of the definitions among the Catalan team).

Actions to bridge gaps included the mutual education of youth practitioners and victim-support services (NGOs). NGOs provided training to improve practitioners' knowledge of support organisations and legislation, and in the process designed materials based on an improved insight into youth practitioners' information needs, and engaged a new audience. The stakeholders for recruitment to the training included the employers (including local government) and professional networks of youth practitioners and, in addition, two experts from Hungary and Serbia: Prof. Dr Judit Takacs and Prof. Dr Vesna Nikolic Ristanovic, respectively, who contributed with research and training experience and helped by checking the translations of the resources and training materials.

This article gives a fuller description of this action project than previously published (although the full funder report has been available throughout, on the project website until 2020 and on the above one since) and offers background to some specific discussions on theoretical, legal or translation-related issues (e.g., Alldred 2023; Alldred and Biglia 2015; Fox and Alldred 2022; guizzo et al. 2017; Jiménez et al. 2016) and highlights the methods and approaches taken to sharing feminist theory with practitioners (see also Biglia et al. 2022).

## 3. Theory and Methods: GAP Work Training in Ireland, Italy, Spain, and the UK

In each of these countries, feminist researchers collaborated with two training organisations to design and deliver the programme, usually a domestic abuse service that was supporting mostly women, and an LGBTQIA+ organisation or a youth work organisation, except in Ireland, where training was brought into a university module on equalities. In addition, they involved other stakeholders for dissemination and recruitment to the training activities.

Youth practitioners undertaking a Master's degree in youth and community work had identified a gap in their knowledge where gendered and/or sexual violence intersects with age, such that if they were confident that they knew how to respond in the case of a child experiencing sexual abuse, they were less certain when a young person was in a consensual relationship or one they considered consensual and experiencing peer abuse. Thus, it was the need to think intersectionally about age and gender and/or sexuality, in particular, and also with race, class, and other structures of privilege that gave rise to the project. Initial theoretical inspiration to think about difference had come from bell hooks (1981) and Audre Lorde (1984) and then later Black feminist scholarship highlighted how feminist work on gender violence had failed to see the effects of racialisation, discrimination, and privilege, and intersectionality was being theorised anew (Brah 1996; Crenshaw 1989; Davis 1981; Phoenix and Pattynama 2006). Phipps (2009) highlighted the hierarchies of class evident in how women who experience rape are treated, and later (Phipps 2020) the privileging of white voices in survivor movements. UK-based sister projects showed the intersections of age with sexuality and gender (for a description see Alldred 2023), and while a common feminist intersectional politics informed each team, in each country race, gender, and sexuality politics took specific forms, such that LGBTQIA+ politics were voiced more loudly in the Spanish and English teams, and ethnic diversity was more racialised in England, and perhaps more coded around faith in Ireland. The Spanish and Italian trainees were less diverse racially than the ones in England, and in the Spanish training sessions, ethnic diversity was mostly post-colonial, with Latin American Spanish speakers the largest minority group. The project team and most of the trainers were white, although diverse regarding gender, sexuality, age, and to some degree class.

Theorisations around gender-related violence(s) and intersectionality have helped question and problematise accepted feminist approaches, as Phipps' (2020) "Me, not you. The trouble with mainstream feminism" attests. "Me, not you" is an (auto)criticism of

heteronormative white feminism. Phipps was one of the partners in the later USVreact Project[2] and calls for challenging our own bias when relying upon and referring to feminist theory. She urges making visible and addressing the inherent 'whiteness' of mainstream feminism, and its exclusivity as well as discrimination against all those who do not fit the mould. In this sense, any educational programme needs to include activities of auto-reflection and auto-criticism, to avoid re-enforcing hegemonic power structures.

Partners shared theoretical and political commitments that motivated the project but were not sure whether youth practitioners needed theory to improve their knowledge and practice or simply updated information about safeguarding procedures and support services. The next section summarises each training programme, trying to characterise its theoretical approach and the way it was delivered and to whom.

### 3.1. Training in Ireland Was Led by the Department of Applied Social Studies at Maynooth University

The Irish 'action', as the funding framed it, was the development of an enhanced equalities training course for youth and community work practitioners in training (during their qualifying degree), and stand-alone training workshops for youth workers already in practice.

Over 200 practitioners, trainee youth workers, and community workers participated in the GAP Work training in Ireland, with 120 completing an evaluation, and 44 identifying as practitioners already, mainly youth workers or community workers. Prior to the courses, students and practitioners participated in focus groups to gather information about their current experience with GBV and their learning needs. They requested more support and training to develop knowledge and skills for intervening and interrupting gender oppression in their work.

The team in Ireland decided to use the term GBV rather than GRV to emphasise that this form of violence is rooted in gender and gender stereotyping. This choice also reflected their greater focus on addressing violence against women and girls (VAWG) compared to tackling homophobia and transphobia. Given the significant presence of migrant organisations and local support organisations, which often have Roman Catholic origins, they felt it provided a more solid foundation to unite all stakeholders.

The training in Ireland was delivered to university students as part of a larger professional programme qualifying them as youth workers, and in a module about diversity, equality and social justice. The materials were not designed to be delivered as stand-alone training, but rather, as part of this larger module. The training provided a feminist conceptual framework for understanding GBV, locating the root causes of GBV within a continuum of sexism, with unconscious bias and casual stereotyping at one end and overt gender oppression and violence at the other end. Furthermore, it placed GBV within the systemic 'vehicle' of patriarchal society that promotes sexist values and practices at personal, cultural, and structural levels.

The training methods approached learning about GBV at the personal level, at the practitioner level, and as trainers on gender violence topics. It was important to work from an understanding that any form of sexism or gender stereotyping dehumanises both women and men and violates women. In Ireland, the GAP Work initiative generated learning for participants and facilitators by building on existing knowledge and resources, developing them further as a result of the prior focus groups, and leaving a sustained project legacy in the degree module that is still being delivered, and in local practitioner knowledge. The evaluation process helped identify gaps and issues when implementing learning in practice. The evaluation research included an anonymous questionnaire, as well as group sessions to receive verbal input and assess knowledge and skills attained. Additionally, trainers (lecturers) held meetings with participants (students) and kept a reflections log and notes of the staff planning and review meetings held.

Youth and community work (YCW) in Ireland and in the UK has often been informed theoretically by critical pedagogy, radical social work, and Marxian social action (Cooper 2018);

thus, the degree programme housing this course would typically contain modules on power and inequalities, and on community work, such that ethical practice with individuals is learned alongside professional obligations towards equality and social justice, an ethics of practice towards the community, not only towards individuals. Degree-level courses do not shy away from theory and the context for the Irish Action therefore differs from that of the others in this respect: in addition to being a component of a larger theoretical body of learning, participants were engaged in long-term relations with their peers and the lecturing staff. Alldred (2017) and Sanjakdar and Yip (2017) have argued that this makes youth work a particularly fruitful framework for learning about gendered violence because individual acts of violence need to be understood in the socio-cultural context of inequalities.

*3.2. Training in Italy Was Led by the Interdisciplinary Centre for Research and Studies of Women and Gender (CIRSDe) at the University of Turin (UNITO)*

The Italian team developed a two-and-a-half-day training course called "GAP Work Italy, Against Gender-Related Violence: Gender Violence Against (and by) Children and Young People: Training for Practitioners". The University of Turin collaborated with two expert training organisations to design the local programme: *GLBTQ*[3] *Maurice Association* and *Demetra: Support and Listening to Victims of Violence Centre* of the Health and Science Agency of the City of Torino University Hospital. Nine training cycles were implemented in 2014: four for professionals from social or educational professions and five for medical healthcare professionals. A total of 210 participants enrolled for 20 hours of training each: two full days of eight hours and a half day of four hours. A total of 182 people participated in at least one training day, and 157 attended the entire course. The sessions took place at the University of Torino to provide a 'neutral' space.

The group of social and educational professionals included teachers in kindergarten, primary and secondary schools, educational services of the Municipality of Turin, youth workers, intercultural mediators, social workers, community helpers, and neighbourhood police who deal with stalking, domestic violence, and give training in schools. Participants from the health professions included nurses, paediatricians, doctors, psychologists, psychiatrists, and a few related students. This was the first and only training the medical staff said they had had on equalities and the team believed it to be the first collaboration between a women's organisation and a 'Pride' organisation in Italy. Other participants included civil servants and politicians who did not work directly with children/young people but wanted to participate to implement anti-gender violence policies or promote awareness-raising activities.

The programme was designed for practitioners working with children and young people on an everyday basis and it aimed to enhance their understanding of sexist, sexualizing, homophobic, violent, controlling, or normative language and behaviour and improve their skills to detect and tackle them. In addition, the training helped build confidence about when and how to intervene and included crucial information about methodologies, tools, and support services. The Italian training was structured as follows:

*Day 1: Accepting differences and questioning norms:* This introductory session focused on various aspects of sexual identity (gender norms, gender identity, and sexual orientation). Trainers engaged participants to analyse and question heteronormativity and gender norms and reflect on the many levels of discrimination and violence targeting LGBT+ people. Trainees were invited to share professional experiences.

*Day 2: Respectful Relationships:* Participants discussed the various forms of violence and its socio-cultural roots, as well as its consequences for people's health. At the same time, the session offered strategies to identify and tackle violence with legal tools, local expertise, and specialist services.

*Day 3: Cascade Support and Evaluation of Training:* This part offered space to review and revisit concepts and material from previous days, discuss specific work-related

needs of participants, prepare for possible interventions in their workplaces, and cascade the materials by sharing their learning with their colleagues.

The Italian training started with an introduction to the concept of gender-related violence and an explanation of how it connected the contents of the three sessions. This bridged the often-present conceptual gap between gender-based violence and LGBT+ discrimination and suggested a broader understanding of violence. It furthermore helped to reflect on the common characteristics of different examples of violence and discrimination analysed during the training programme, underlining the influence of heteronormativity and gender norms.

### 3.3. Training in Spain Was Led by the University Rovira i Virgili (URV)

The Spanish training was implemented across Catalonia and published in a guide called *Joves, Gènere i Violències. Fent nostra la prevenció* (Youth, Gender, and Violence: Gaining Agency in Prevention) (Biglia and Jiménez 2015). It consisted of five five-hour sessions, an online tutorial, and a half-day for evaluation.

The training was led by URV and delivered by two non-profit feminist associations with expert trainers: *Candela*, which promotes social transformation from a community perspective and works on GRV prevention through education on sexualities, and *Tamaia*, which works on violence against women and offers support to survivors.

A total of 200 people were recruited for the training, primarily through the Catalan Department of Education and the Catalan Youth Agency. In the end, 189 professionals attended at least one session, and 164 attended at least 80% of the sessions. The majority of participants were women (84%), were born in Catalonia, had advanced studies (56% undergraduate, 35% Master's or PhD level), and had an average age of 40.5 years. The groups were quite homogeneous, consisting of highly educated, non-migrant women, and were representative of professionals in Catalonia. Of these, 61% of them had previously participated in GRV training, and 6% had studied a Master's or other specialised programme.

The team in Spain designed an innovative course by focusing on the structural and heteropatriarchal roots of gender violence, highlighting that it is not a personal issue between two individuals (often assumed to be a man and a woman), but a cultural one. In this respect, their training programme can help education and youth work practitioners to be more aware of social norms and more respectful of individual and cultural differences when intervening.

The following table (Table 1), taken from the GAP Work Project report, summarises the contents and learning outcomes for each of the five training sessions:

**Table 1.** Outline of the training by the GAP Work Project team in Spain.

| Session and Title | Contents | Learning Outcomes |
|---|---|---|
| A: Introduction to the roots of 'gender-related violences' | Understand that the roots of GRV are socially constructed and reproduced and that we internalise them, especially during childhood and adolescence. | Develop a personal sensitivity to normativity and GRV that allows self-review of professional activities and personal experiences. |
| B: Abuse, control and violence in intimate relationships among young people | Understand the complexity of macho violence in sexual or romantic relationships between young people and their links with ideas about romantic love. | Be able to produce a safe environment that allows early detection of abusive and controlling relationship dynamics. |

**Table 1.** *Cont.*

| Session and Title | Contents | Learning Outcomes |
|---|---|---|
| C: Violence related to gender identity and sexual diversity in young people | Understand the complexity of the dynamics of discrimination and the violence against non-heterosexual people and its effects on the comprehensive education of young people. | Be able to generate a respectful environment towards sexual and gender diversity that favours the detection of abusive dynamics. |
| D: Prevention of GRV as a key tool for its eradication | Be aware of the influence of different socializing agents for both maintaining and eradicating different expressions of GRV. | Develop a creative and motivating attitude to the prevention of different forms of GRV among young people. |
| E: Let us put it into practice! | Learn how to implement and evaluate the knowledge acquired. | Enhance the transformative capacity of the professionals who work with youth. |

In accordance with feminist thinking, the programme not only focused on theory but actively encouraged participants to self-reflect on internalised gender stereotypes and relate the activities to their own experiences (Giraldo and Colyar 2012). The training aimed to raise awareness, foster critical consciousness, and inspire participants to actively transform social norms (Rebollo-Catalán et al. 2011). The programme in Catalonia served as a political intervention, providing participants with an opportunity to engage in a personal and collective journey that drives social change (Mayberry 2001). It adopted a more reflective 'consciousness-raising' approach than the others and the repeated meetings of the same group of professionals over five days allowed trust to be developed within the group. The discussion of intersectionality was wide and theoretical and was not met with resistance, despite less diversity within the trainees' groups themselves. Some videos were made to share elements of their experience and they offer useful resources for later cohorts of professionals and those who wish to be inspired by the changes in practice the trainees committed to.

*3.4. Training in the UK Was Led by Brunel University London*

In the UK, the team conducted a three-day education programme in England that was condensed into two-and-a-half days at the request of an employer of a large group of the participants (local government). Each of the training days had a specific focus and a different expert facilitator. The sessions explored gender, inequalities, and violence in various contexts in young people's lives, and also reflected on violence and discrimination in the workplace in response to reflections offered by some of the practitioners. The first day's training focused on defining and identifying gender-related violence, and on examining the values and norms structuring the social contexts in which it occurs. In this respect, it had similar politics and theory to the Spanish training and its inspiration from the Nordic norm criticality work within sex education is described in Alldred (2023).

Day 2 provided information on the law (sexual consent, sexual offences, domestic abuse, and hate crime) and legal aspects of response and offered guidance on promoting positive relationships for working directly with young people and support to discuss sexual issues with them. The overall programme emphasised the need to look critically at norms and expectations, included activities to promote participants' reflections on their own assumptions, and highlighted actionable steps that trainees could take in the future by asking them to create an action plan each session (see Cooper-Levitan and Alldred 2022 for analysis of these). Day 3 focused on taking interventions in the workplace, cascading the learning to colleagues, and on recognising dynamics of power and domination.

The UK local actions were coordinated by the Centre for Youth Work Studies (CYWS) at Brunel University London. The programme was implemented in partnership with a national legal advice and policy organisation, *Rights of Women,* and a youth work training organisation, *About Young People.* The sessions took place at various venues, including

the Initial Teacher Education Department at Brunel University, the London Borough of Lewisham, the Institute of Education, and Coventry University, all of them in England.

A total of 180 people registered for the training and ultimately 156 individuals attended the first day, and 128 successfully completed the three-day training. This number was disappointing and apologies usually cited 'pressure of work' as an explanation for not attending all three sessions, and indeed the team did hear of incidents among young people that drew the attention of the community safety team away from the training one day. Participants included teachers in training, practising youth workers, and other professionals. The largest group received training in Coventry, which led the organisers to believe that there is a greater demand for such training outside of London, where people may have numerous options.

The UK training was designed with the following objectives in mind (see Table 2) copied from the GAP Work Project report):

**Table 2.** Outline of the training by the GAP Work Project team in the UK.

| Overall Aims | Objectives | Outputs | Outcomes |
|---|---|---|---|
| To enable youth practi-tioners to: (a) Recognise gen-der-related violence (GRV) in their set-tings; (b) Confidently inter-vene and take action to combat GRV; (c) Refer to appropriate agencies; (d) Share their learning with colleagues. | To educate participants on the nature of gender-related violence (GRV). | 3 training workshops per cohort. | By the end of the programme, par-ticipants will have: |
| | | 3 action plans per participant. | Ability to reflect on, and to challenge personal values, attitudes, and experiences. |
| | To train participants to recognise GRV and refer to appropriate agencies. | 1 resource pack with hand-outs and relevant information for each training day. | Gained knowledge, skills, and re-sources to recognise and identify GRV. |
| | To enable participants to 'cascade' their learning to others (e.g., col-leagues). | 1 'Cascade' resource pack. | |
| | | 2 'legacy' documents. | Gained motivation and confidence to take proactive steps to prevent and react to GRV. |

The training was designed as an initial introduction for professionals who might encounter homophobic and/or sexist bullying in settings such as playgrounds, sports grounds or youth clubs. However, it was open to adaptation by trainers to suit their specific contexts. It did share theory with participants explicitly, especially on the first day, and feedback from the survey at the end of each day suggested that the discussion of the social construction of gender was a bit too theoretical for some practitioners, and, like the theoretical discussion of GRV itself as wide, inclusive, and norm critical, it was sometimes welcomed as generating insight by linking issues together and sometimes not wanted (Cooper-Levitan and Alldred 2022 discusses this further). This training addressed participants as professionals and asked them to reflect on their work with children or young people and did not ask them to share personal illustrations of the violence or inequalities discussed. This approach was only agreed upon after much discussion among the UK team and reflection on the cultural differences among them. The decision about the use of feminist and intersectional theory was less controversial than this question about how personal to be. Pedagogic practice informed the activities which involved deriving learning and applying principles themselves. Although the overall programme was jointly written, different facilitators led each day, so a feminist lawyer led the rights session (day 2) and a youth work practitioner led the day that focused most on action planning (day 3), and the trainer for day 1's discussion of social theory and gender had a PhD and was a lecturer and an experienced youth and community worker. The coordinator and this trainer have reflected on the use of pedagogies of discomfort for learning about such topics (Cullen and Whelan 2021). Approval and certification by a national teachers' union consolidated the status of the training, but prevented further adaption beyond this point.

## 4. (Feminist) Theory into Practice

### 4.1. Feminist Pedagogies

Each of the four original programmes (training courses) had a somewhat reflexive approach to learning about gender, and used personal experience to identify and make conscious cultural norms. However, the extent to which the training programmes used personal experience to derive insights or used social pedagogy (see e.g., Hatton 2018) or Freirean critical pedagogy varied and they can be seen as occupying a range of positions on a continuum from using subjective experience to using 'proven knowledge' to motivate and inform practitioners. The Spanish programme was the most reflexive and employed pedagogies linked to feminist practice in an education department. The UK and Spanish teams have written about their approaches since then (see Biglia and Cubero 2022; Biglia and San Marti 2007; Biglia et al. 2014; Cooper-Levitan and Alldred 2022; Cullen and Whelan 2021). Cullen and Whelan (2021) have written about how pedagogies of vulnerability may be applied and the UK co-trainer and researcher who sometimes swapped roles, reflected on the way some trainees responded (poorly) to their gender/gender presentation and age.

### 4.2. Issues of Translation

From the outset of the project, the term gender-related violence (GRV) played a central role in fostering discussions on how to develop training from an intersectional perspective. In addition to its novelty, GRV faced challenges with translation (see guizzo et al. 2017). To ensure a contextualised response that aligns with local cultural norms and language specificities, each partner translated or utilised the term in distinct ways.

In Italy, the team employed the term GRV as defined within the project, encompassing "sexist, sexualizing, or norm-driven bullying and harassment". Their intention was to design an innovative programme that addressed issues not typically included in discrimination and violence prevention training in Italy. While the city of Turin has a longstanding tradition of training on VAW, there has been less emphasis on gender identities and sexual orientation, especially for children. Moreover, these subjects had not previously been integrated into a single programme.

Over the years, the terms "violenza maschile" (male violence) and "violenza di genere" (gender violence) have primarily been used in the context of "violenza contro le donne" (violence against women), particularly by feminists, the mass media, and local policies. However, these terms are often misused. For example, "gender violence" is sometimes used interchangeably with domestic violence (DV) or violence against women (VAW). Such misuse demonstrates how gender-related issues are still equated with women's issues, limiting a broader understanding of the phenomena and hindering men from disclosing sexual, domestic, or homophobic abuse.

To counteract this tendency, the team in Italy paired trainers with two different backgrounds: experts in LGBTQ+ rights and experts in the area of VAW. This approach allowed for the design of a programme that introduces various themes and bridges the gaps between them.

In Catalonia, Spain, the team adopted a comprehensive definition of GRV by using the plural form: "violencias de género" or gender violences (Biglia and San Marti 2007). The rationale behind this choice is explained in the final report:

> "This approach aims to highlight that gender itself is a form of violence, as it compels individuals to conform to a predefined, dichotomous construction of identity. When referring to 'gender violences', we include all forms of violence that occur and are reproduced within gender relations and social roles. The sex or gender of the subject perpetrating or experiencing the violence is therefore irrelevant since even an ungendered body or institution can exercise or experience it. The interconnectedness between the construction of gender and the heterosexual imperative means that violence against LGBT individuals is also considered a manifestation of gender violence. The focus is on a comprehensive understanding of violence that encompasses power dynamics within relationships, lesbo/homophobia/transphobia,

and violence manifested through institutional, symbolic, and community relations. However, while different forms of gender violence share common roots, they are not equivalent and may not necessarily produce the same emotional effects. Consequently, understanding their causes, processes, and particularly their effects is crucial. An intersectional approach is essential since gender violences must be understood within the context of the embodied subject, experiencing them in specific socio-cultural contexts" (Alldred et al. 2015).

These reflections on "violencias de género" aided the Spanish team in developing a coherent training focus. They recognised both the advantages and potential pitfalls of trainers with different backgrounds, as their approaches could potentially create a disjointed programme that leads participants to view violence against LGBTQ+ individuals and gender violence in partner relationships as separate issues. However, the evaluation of the training course showed positive results in this regard: 15% of participants reported a significant increase in their ability to recognise gender norms. This likely reflects the training's emphasis on the importance of heteronormativity in the construction of gender.

*4.3. Definition Decisions*

The training material designed by the Spanish team emphasised that gender violences have heteropatriarchal and structural foundations. In this regard, the Spanish and UK programs shared theoretical similarities. However, the Spanish programme had a more personal approach that encouraged trainees to reflect on their own experiences. In contrast, the UK programme focused more on the professional context, but both programs aimed to support professionals who work directly with children and young people, encouraging them to approach interventions with an intersectional perspective and respect for differences.

In Ireland, the team preferred the term gender-based violence (GBV) in preference to gender-related violence (GRV) used by the other partners. They believed that the term GRV could potentially be confusing to trainees, because it might not explicitly convey the underlying connection between gender and violence. Using the term GRV might lead trainees to perceive gender as merely tangentially related to the issue of violence, rather than recognizing it as a central factor in the perpetuation of violence. By choosing the term gender-based violence, particularly for the in-practice trainees, the team emphasised the fundamental role of gender in the occurrence and perpetuation of violence. They wanted to ensure that trainees understood that GBV is not simply violence that happens to be associated with gender, but rather violence that is deeply rooted in gender dynamics, gender stereotypes and societal norms. By using the term GBV, they hoped to provide a clearer and more focused understanding of the issue, allowing trainees to recognise the importance of addressing gender norms and promoting gender equality in efforts to prevent and respond to violence.

The Spanish and Italian teams chose GRV to encompass a broader range of violence that is connected to gender, including not only violence against women but also violence targeting LGBTQ+ individuals and other forms of gender-based discrimination. Their approach highlighted the interconnectedness of forms of violence and discrimination related to gender, challenging heteronormativity, and promoting a comprehensive understanding of gender violence. Overall, while the Spanish and Italian teams took a broader approach to encompass various forms of gender-related violence(s) and discrimination, the Irish team aimed for a more explicit emphasis on the role of gender in violence, highlighting the need to address gender norms and promote gender equality.

The UK team favoured a broad definition of gender-related violence. They identified three main themes within this definition: violence against women and children, violence based on homophobia and transphobia, and violence based on 'machismo', which includes violence from men with hegemonic masculinities against other men. The trainers maintained flexibility in their approach, focusing more on one theme or another based on the needs of each group or example, regarding personal, group, institutional, or professional

needs. This allowed them to address emerging issues during training and even identify them tentatively beforehand during recruitment. Some theoretical tensions arose during the delivery of the UK training. The main themes of the sessions were selected not only based on the trainees' needs but also on the trainers' preferences. For instance, the training partner Rights of Women, an organisation rooted in second-wave feminism, sought to prioritise the legal aspects of VAW. This occasionally led to a hierarchical arrangement of themes, with VAWG placed above general 'machismo'. To counteract this hierarchy, activities promoting healthy relationships and for discussion of relationships with young people were incorporated into the programme (Rights of Women 2014). To counter heteronormative tendencies in workshop discussions, the training materials included same-sex relationships and a non-binary character in the vignettes.

While the different terminology used by facilitators to explain GRV reflected politicised positions in response to local contexts, these discrepancies did not have a negative impact on the training. Final evaluations across the countries demonstrated that trainees felt they had gained a better understanding of GRV and expressed confidence in their ability to prevent and intervene in GRV situations.

The next section highlights lessons about training methodologies.

## 5. Findings from the Four GAP Work Project Actions

Following the logic of the GAP Work Project's separate and specific local actions, the findings are presented by the team.

In Ireland, there was a strong interest and need among youth workers to explore issues related to gender, identity, and overall equality. The trainees recognised that as youth workers, they were in an ideal position to support young people navigating their place in society and challenging oppressive norms, such as heteronormativity and patriarchy and for LGBTQI+ individuals and young women to assert their identities. Self-exploration and gender consciousness, integral parts of the training, are crucial for youth workers who serve as role models for young people.

The Irish team acknowledged that exploring personal experiences during the training was intense and required adequate space and time for reflection and closure after each session. It was important to prepare trainees for the content of the training, as it could be emotionally challenging. Trainees felt more comfortable in separate gender identity groups, where they could open up and identify with others of the same gender. Within each group, at least one person disclosed experiencing sexual abuse as a child. Trainers need to be well-prepared to support those who disclosed and provide support to the entire group. Initially, some men found it difficult not to feel accused or defensive, while some gay men felt uncomfortable discussing sexism and homophobia in a male-only group. The separation of groups by gender allowed for discussions on heteronormative dominance but also raised contradictions and potential offensiveness for gender-fluid, non-binary, or undisclosed individuals.

The Italian team reflected on good practices in GRV training. They found that LGBTQ+-related themes were often overlooked, but the GAP Work training successfully addressed these themes by challenging binary, heteronormative, and heterosexist perspectives on gender. The training stimulated reflection and provided valuable tools, including knowledge about local services and effective anti-violence strategies. The project also facilitated networking among trainers and strengthened the local support network. The training methods employed, such as individual and group exercises, classroom discussions, and opinion exchanges with trainers, had a significantly positive impact. They provided support for professionals to share and confront workplace difficulties and fostered reflexivity, allowing past workplace incidents to be reconsidered and personal experiences to be viewed in a new light. For instance, participants recognised that they or their colleagues had made assumptions about the heterosexuality of clients or colleagues.

On a structural level, the concept of intersectionality was a valuable analytical tool to understand how GRV is normalised and how inaction perpetuates it. However, participants

felt overwhelmed when they realised the pervasiveness of sexism and GRV. To combat this, the training focused on identifying concrete ways in which practitioners can intervene, interrupt, and combat sexism in their everyday lives and work, but unlike youth professionals, health staff had less chance to focus on this within their specific and medically responsive role. It was noted that there is limited work being done in youth work contexts regarding gender roles and masculinity with young men, or the gendered experiences of young women. More emphasis should be placed on developing respectful and healthy relationships beyond sex education programs, where it is usually addressed.

The use of specific examples and case studies proved highly useful in understanding real-life situations. However, participants often sought universal solutions that could be applied to any situation and trainers needed to remind participants that there are no ready-made solutions or remedies that apply universally, but rather principles to apply and tools to adapt to each unique case. This supported the approach of 'education' (values, principles, theoretical models to apply in situ) rather than training (meaning to respond in particular ways), but delivered via what are common training methodologies (in a topic-focused, group learning environment away from the desk).

The training needs to cater to the requirements of different professional trainee groups. Legal aspects were discussed in relation to the specific occupations of the participants. Medical and healthcare professionals requested more specialist information on the effects of GRV on people's health and specific provision for intersex individuals, while also reflecting on the social construction of sex and gender. Participants across various groups expressed a desire for more focus on legal aspects. However, it is worth considering whether this is an actual training need for professionals not working in a legal or judicial environment or if it is a perceived need influenced by the context of increasing legal actions against healthcare personnel, causing heightened anxiety. This context has also diminished the authority of teachers in the eyes of families, creating uncertainty about their responsibilities. The law was sometimes sought for clarity and certainty, which it could not always provide, itself being a framework applied in a particular context where the details matter.

The Italian and UK teams expressed concerns about budget cuts to public and health services, which limit the ability of staff to tackle GRV within their organisations. Furthermore, integrating new learning into work practice poses ordinary challenges and highlights the need for organisational change, clear commitment from senior managers, and general education for mutual respect. For most participants, this training was an additional, usually optional, component, although in Ireland, was a part of the accredited youth and community degree course. It is important to mainstream this topic into the curriculum of all professional education courses.

In Catalonia, Spain, the partners used feminist pedagogy to create opportunities for participants to critically analyse their experiences through gaining a sense of distance. This process was time-consuming, and participants consistently requested an increase in the course duration. They expressed satisfaction with the exercises and resources and reported enhanced confidence and awareness regarding GRV and intervention. Participants desired more case studies and problem-solving materials, suggesting a need for smaller groups to facilitate these activities. Alternatively, a two-stage programme with an initial focus on understanding GRV and a subsequent stage emphasizing intervention could be beneficial. Trainers with prior expertise in the course topic were essential, and Spanish trainers received high evaluations from participants. However, the materials developed by trainers during the project would need to be adapted for use by others in the future, considering local contextual factors and intersections. Participants and trainers need to confront their own sexist, racist, or homophobic prejudices. Additionally, more emphasis should be placed on understanding the impact of intersectionality in GRV. The Spanish team emphasised the importance of focusing on intersectionality and suggested further specific research in this area, and the UK team urged consideration of intersectionality in the reception of GRV training.

In the UK, the training programme demonstrated positive outcomes for participants, both quantitatively and qualitatively. Attendees felt that they had acquired new knowledge, understanding, and skills to address the themes of GRV explored in the training. They found the legal aspect of day 2 particularly useful. They applied the theoretical concept in areas not covered by the vignettes such as in police behaviour toward young people or staff behaviour (such as heteronormative assumptions) at work. However, it was hard to know if recognition of power and inequalities had been effectively applied in practice by all the UK participants. Trainers and researchers sometimes had concerns about the varying interpretations of the concept of GRV, influenced by different philosophical positions and professional experiences. This risked leading to theoretical incongruence. Studying the intended interventions back in the workplace and whether they achieved legibility in the work context would give a valuable reflection on the training (Cooper-Levitan, forthcoming). The role of location, space, and positionality was underestimated, particularly concerning faith-based work settings. Participants in these settings face the complexity of introducing 'secular' feminism and critical pedagogy to orthodox faith leaders, often heterosexual males resistant to certain GRV themes. These participants need more intensive support to navigate the path of embedding the training into their practice.

Moreover, it remains to be determined whether a feminist/critical praxis can be achieved through short bursts of continuing professional development. Participants highlighted that implementing the necessary changes to address GRV required support from managers and strategic input. Many senior managers were notably absent from the training sessions even where whole teams were booked in. Observations revealed participants' concerns about receiving support from organisations/institutions and the impact of professional 'audit' cultures on forming a critical praxis. Obstacles such as a shortened school day and a narrowing curriculum were identified by trainee teachers, while others pointed out the lack of resources in the voluntary sector where much youth work occurs. One trainer expressed concerns about opening a 'Pandora's box' and emphasised the need for supervision and developmental support to ensure meaningful application of the training.

## 6. Recommendations for Training on Gender and/or Violence

The four training programmes were designed and piloted during the GAP Work Project with different political and pedagogic logics given their various cultural contexts. While a more detailed discussion of recommendations can be found in the project's final report (Alldred et al. 2015), some overarching recommendations regarding training are likely relevant to many training/educational interventions on gender or on violence:

- Locate violence within the broader context of inequalities.
- Recognise the intersectionality of structural inequalities and cultural exclusions, particularly in relation to race, ethnicity, age, sexual orientation, and class, both in the topic and among the participants.
- Develop training programs that are grounded in the concept of GRV or gender-based violence (GBV) to maintain theoretical coherence.
- Encourage trust, support, and confidence in critically reflecting on practice, by making explicit that poor practice stems from social values and norms, and can be addressed culturally rather than by blaming individuals.
- Enable reflections on workplace dynamics and staff experiences, as well as service users' experiences of violence, to reduce the 'them and us' dynamic and also address the potential risks for trainees (e.g., informing them in advance if their work colleagues will be part of the training group).
- Question what is considered violence and challenge the normalisation of problematic behaviours.
- Ensure trainers have information available for individuals seeking further support on the issues raised, and acknowledge the potential personal impact of the training.
- Employ diverse training approaches to accommodate different personal connections to the topic, such as using distancing techniques or facilitating separate male and

female-identified discussion groups to encourage the sharing of personal experiences if this is what trainees want.

- Allow enough time for the processing of the material personally as well as intellectually, and the staffing ratios.
- Our recommendations on training group size are that two trainers (or a trainer and a co-facilitator) were necessary for a group of 20 trainees in order to be able to respond adequately to personal realisations and disclosures (Final report p92).
- Strike a balance between conveying hope and addressing the despair associated with violence or abuse.
- Avoid judgmental attitudes and promote personal reflexivity in training, recognizing that social change is needed to address heterosexist, sexist, and racist environments that perpetuate GRV.
- Ensure this open approach extends to the trainers who can model being reflexive about their positionality, biases, or own learning journeys.
- Allow enough time for handover between trainers to include a report on the group dynamics and learning so that the later trainers benefit from the insight gathered and can build on what has been achieved so far. We found having a table to complete or a sheet with prompts helpful, as well as an in-person debrief.
- Where multiple trainers or lecturers contribute within a programme consider whether the same vignette or example can visited by them to add layers of learning, or make clear how later points build on earlier ones.
- Reflect upon what differences may structure or nuance the relationship with the trainers and consider the value of discussing these explicitly in the session.
- Do not underestimate the importance of establishing ground rules for all at the outset, even if these feel very familiar, as it is part of the preparation of trainees emotionally and signals what is expected behaviourally, and because commitment to them can be tested (ibid. p. 92).
- We recommend that thorough preparation of trainees for the session includes an explicit statement about the topic, type of activity, and discussion style so they arrive knowing the potential emotional impact and degree of sharing invited and can make decisions about participation accordingly.
- Participants should voluntarily sign up for the training with full knowledge of the topic and the potential personal and professional risks involved.

We found that the composition of training groups, in the sense of comprising one professional group or diverse professionals in the session can produce variable consequences. Having a mix of professionals allows practitioners to learn from each other, while more specialist training can be provided to particular professional groups. Both options are valuable, but it is important to align the training approach and pedagogy with the professional status of the participants. It is recommended to group professionals from diverse fields or those who share a similar degree of professional autonomy or status as within such peer groups, participants can exchange ideas on shaping practice and interventions at a comparable level. It would seem unfair to discuss interventions such as the youth workers' Action Plans with those health professionals whose role does not usually allow such innovation.

Many of these recommendations may seem like common sense to experienced trainers in GRV issues, but they are crucial to remember when organising new training programmes. These points should be defended when seeking funding or implementing training. It has been observed by GAP Work partners that GRV training is often limited to a basic session included as a tick-the-box element in equality training, while more comprehensive GRV sessions would benefit all professionals. Moreover, small group sizes, ample preparation time for both trainers and trainees, and relevant post-training follow-ups are not just preferred but are essential to create a safe and effective intervention. However, these elements require time and resources, which are often considered excessive. With these recommendations, we aim to support the argument that GRV training is essential for the

professional development of youth practitioners and teachers and professionals in a range of roles, since social change is needed at the broadest level.

## 7. GRV and Intersectionality beyond the GAP Work Project

We believe that there are important practical applications of feminist theory and critical pedagogy and we hope that the lessons learned from this project may be applied to other forms of difference and marginalisation that intersectionality theory highlights. Our recognition of our own partiality and specific formation as trainers, lecturers, or researchers is important in order to identify our projections and assumptions about learners.

The partners of the GAP Work Project have continued exploring gender-related violence(s) and intersectionality. In particular the USVreact Project - Universities Supporting Victims of Sexual Violence: Training for Sustainable Services—applied lessons from GAP about the development and delivery of training and focused on the issue of sexual violence. It sought to improve response to sexual violence (SV) at universities, at the level of policy and care pathways, and at the 'first responder' level, so provided training for people who may have initial contact with survivors and need to be prepared to support them appropriately and refer to specialised services. This EU co-funded project, led by Brunel University London had seven partners and many associate partners across six European countries and between 2016 and 2018 developed seven innovative training programmes for university staff who may receive disclosures of sexual violence, and sought to embed these within their institutions.[4] It took forward the broad definition of gender-related violence which meant it emphasised an inclusive approach to sexual violence disclosure that urged the recognition of how intersecting differences (race, sexuality, class, status, age, disability, etc.) can shape and hinder disclosure experiences and interventions (Page et al. 2019). Partners knew that despite designing a shared theoretical framework, based on shared concerns, they needed to design different training programmes and material for each local context, responding to different needs and this time it included care pathways and response protocols, processes for organisations to follow within their differing legal and social contexts. They had learned during the GAP Work Project that they needed to remain flexible when designing training for each specific context and target group.

Whilst piloting USVreact Project training across different universities in their country, partners found different needs, pre-conceptions and understandings of terms, situations and thus solutions. For example, interventions at a small university in a rural setting with a tight-knit community are not like interventions at a large, urban university with students living off campus whose sense of community is more diffuse. The broad definition of GRV seemed particularly important in the UK given renewed evidence of the prevalence of violence against LGBTQI+ and non-binary students (e.g., Bull and Turner-McIntyre 2023), but data is needed on violence in other countries. The novelty here was to include not only university teachers and tutors, but also security staff, library staff, management staff, and student association leaders, so that anyone who might receive a disclosure was prepared to respond well. The range of training programme models and approaches is evident in those available for adaption and use and includes specific courses for departmental managers who may be supporting their staff who receive disclosures [www.usvreat/eu].

In another project, ACTIVproject[5], about supporting women survivors of domestic abuse on their path back to work, partners decided to explain on the website and alongside the online resources the added complexities to the construction of gender through language. Here particularly the use of the term 'domestic violence' in some languages (French, Romanian) caused a problem in Spanish and Catalan, where the term is no longer used (as explained earlier), but instead "violencia de género" and "violencia de gènere" is used in the vernacular as well as the law. The challenges of linguistic and cultural translation on these topics are explored further by guizzo et al. (2017).

We found that the term GRV supported thinking intersectionality. For the USVreact Project, an intersectional perspective was central to designing inclusive training and responses that respected the experience of all survivors of violence. One of the partners,

the University of York, published the "Guide for Training Facilitators: Developing an Intersectional Approach to Training on Sexual Harassment, Violence and Hate Crimes" (Page et al. 2019). This accessible document was developed from focus group research with key figures, including student liberation officers and university staff. Their discussions focused on differences between individuals and their proximity to and risk of experiencing violence, and how this might influence their ability to disclose or to intervene safely. The booklet offers guidance to training facilitators on how to incorporate intersectionality into existing training on bystander intervention and first response to disclosures of violence. It is designed to enhance existing training by explaining what intersectionality is and suggests how to frame training from an intersectional perspective, as well as particular activities facilitators can use. It again emphasises that there is no one way to design and implement GBV/GRV training.

Training approaches that take intersectionality as its underlying theory make practical use of the theoretical work. One example of good practice is the project "Igualtats connectades: Intersectionality in local public policies". Another EU co-funded project implemented between 2018 and 2019 in Terrassa (Spain) investigated the possibilities of applying the principle of equality and non-discrimination in an intersectional way, in the context of a local authority. The project organised a series of training activities on intersectionality, including five workshops for council officers (over 100 participated) and a series of debates on intersectionality, aimed at professionals working in equality policies, companies and organisations in the city, which more than 290 attended. The project produced free resources, such as the "Toolkit to incorporate intersectionality into local policies" and the "Reflections for developing plans, programmes and projects from an intersectional perspective" (Coll-Planas and Solà-Morales 2019) as well as 10 videos of both the *training sessions* and the debates in the city.[6] The project activities provide examples of how to apply an intersectional perspective when organising a training programme at the local government level. It involved public servants and civil society and focused on a range of cross-sectorial topics. Gender remained one central subject to the training interventions, alongside others such as origin or ethnicity.

To address our initial questions—is it helpful to share feminist theory with youth practitioners and is there room for it on short training courses such as in EU 'Action Projects'?—our response overall is yes, but qualified as above. These qualifiers are important: theory needs to be presented accessibly and applied well, to demonstrate its value not assume it, or else it risks alienating participants and reinforcing divisions between those with an education or not. Critical pedagogy and coproduction approaches start from the values and concerns of participants rather than importing theory, so these can work well for some groups, but sometimes it can help raise the status of an issue to share some of the theoretical work that has been done on it. In response to the question of whether theoretical work on intersectionality and the concept of gender-related violence (GRV) which grew from it can be shared in training interventions with professionals in everyday contact with children and young people, the conclusions are more mixed. The UK team had the most positive findings on this, with youth practitioners reporting making links that we theorists understand through the concept of intersectionality: for instance, seeing links between homophobia and domestic abuse experienced by women where they had not previously drawn a connection, and applying their values and behavioural expectations of young people to their colleagues and to the police generated new insights and critical awareness about norms and also the unnoticed violence of misogynistic lyrics which they left inspired to discuss with young people (Cooper-Levitan and Alldred 2022). However, where feminist theory was less explicit in the training, it informed the understanding presented by trainers and so we would argue that practitioners were using intersectionality to improve their practice by their consideration of forms of difference and privilege in vignettes and examples. Training on gender-related violence has to be designed with an intersectional perspective in order to reserve space to make visible underlying power structures specific to each topic, setting, and training group. For instance, the most difficult form of power to

discuss in the Italian training might have been that coming from the hierachy of medical over other (e.g., care) staff in a hospital context. This illustrates the value of having the theoretical tool to apply in situ since trainers might not have arrived with deconstructing medical authority on their agenda yet it might have been significant to particular cohorts of trainees. Feminist work on violence, inequality, and intersectionality and forms of critical pedagogy make good sources of theory for feminist research and education about violence (Luna and Rubio-Martín 2022), and as a reviewer kindly shared, Bal (2012) highlights the value of concepts and metaphors, rather than whole theories as 'sites of debate, awareness of difference, and tentative exchange' which is exactly what we seek for our training both 'in the room' and afterwards 'in the academy' reflecting on different partner experiences.

**Author Contributions:** Writing—original draft preparation, g.g.; writing—adding sections, revising and editing, P.A. All authors have read and agreed to the published version of the manuscript.

**Funding:** This research was co-funded by the European Union's Daphne-III Programme (JUST2012/DAP/ AG/3176) and led by Pam Alldred at Brunel University London (UK) between 2013–2015.

**Institutional Review Board Statement:** The study was conducted in accordance with the Declaration of Helsinki, and approved (UK action, overall project) by the Brunel University London (UK) (School of Health and Social Care) Research Ethics Committee on 13 January 2013, which also gathered the Maynooth University Ethics Committee approval and statements about local practice from the Italian and Spanish Local Action Coordinators. www.brunel.ac.uk/people/project101293 (accessed on 1 January 2022).

**Informed Consent Statement:** Informed consent to study their learning and feedback was obtained from participants in exchange for free training, and in Italy and Spain the study complied with local expectations of good practice at the time which did not involve formal research ethics approval process.

**Data Availability Statement:** The study did not gain permission to archive the data. Contact the Local Action Coordinator in each country (see report on USVreact website for their names/emails) to make specific requests.

**Conflicts of Interest:** The authors declare no conflict of interest.

## Notes

[1] Due credit should be given here to all the contributors to the GAP Work Project final report, who were, in Spain: Barbara Biglia (Coordinator and legal analysis), Edurne Jiménez (local action & evaluation), Pilar Folgueiras (evaluation), Maria Olivella and Sara Cagliero (legal analysis), and Anna Velasco and Jokin Azpiazu (survey analysis); in Italy: Chiara Marcella Inaudi; in Ireland: Berny McMahon and Oonagh McArdle (Coordinator); in the UK: Fiona Cullen (Coordinator), gigi guizzo, Malin Elge (nee Stenstrom), Mika Neil Cooper-Levitan, and Ian Rivers. Miriam E. David and Ian Rivers were Project CoIs, and Miriam co-edited the final report with Pam Alldred.

[2] USVreact Project: Universities Supporting Victims of Sexual Violence: Training for Sustainable Services. Available online: https://usvreact.eu/ (accessed on 22 December 2022).

[3] Gay, Lesbian, Bisexual, Transgender, Queer.

[4] Universities Supporting Victims of Sexual Violence: Training for Sustainable Services' (USVreact), project code: JUST/2014/RDAP/ AG/VICT/7401, duration: March 2016–2018, was produced with the financial support of the DG Justice: Fundamental Rights and Citizenship Programme of the European Union.

[5] ACTIVproject https://activproject.eu/resources/definitions/ (accessed on 1 November 2022).

[6] Igualtats Connecatdes Training videos 2019 https://igualtatsconnect.cat/en/training-resources/ (accessed on 1 November 2022).

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
