# Peer review of "Tackling Gender-Related Violence: How Can Theory Inform International Professional Education Projects?"

_socsci, doi:10.3390/socsci13010061_

Round 1

Reviewer 1 Report

Comments and Suggestions for Authors

I have carefully reviewed this manuscript and below is my decision.

- The paper is very simple. Written like student homework. Since it is not an application, it does not contribute to the literature. As such, it cannot be considered a scientific study.

The current work is not suitable for publication.

Author Response

Dear reviewer, 

thank you for your time in reading our paper and giving detailed feedback. Please find attached our responses to yours as well as the other reviewers comments. 

With kind regards

Reviewer 2 Report

Comments and Suggestions for Authors Standardisation of the information provided for each research site would be helpful, or noting that incomplete information is a result of data not being gathered. Some of this information could be usefully summarised in (a) table(s). So far these sections contain Ireland - training cohort size - Evaluation cohort size - Professional roles of those in evaluation cohort - Broad content of training - Definition of GBV/GRV used Italy - Length of training - Training cohort size - Professional roles of those in evaluation cohort - Detailed content - Definition of GBV/GRV used - Delivery partners and their role Spain - Length of training - Delivery partners and their role - Training cohort size - Training cohort demographics - Detailed content - Theoretical framework used UK - Detailed content - Definition of GBV/GRV used - Delivery partners and their role - Training cohort size * In the Lessons Learned section, the paragraph beginning "Finally..." may be more useful at the start of the section to frame what follows. * on p13 the discussion of USVreact (whilst interested) doesn't seem to be as contextualised as other sections, the relevance needs to be made clearer. * I wondered whether Bal's (2002) concept of 'travelling concepts' might be helpful in the translation section - this is a really interesting section but lacks reference to the exte siv * The structure may benefit from a review to focus the introduction and early sections on the analytical points rather than starting with a very descriptive section without a clear sense of the argument being made. * Overall, the piece would benefit from a thorough edit and proofread and a more coherent conclusion. It's also not entirely clear why the conclusion is introducing new discussion (of Phipp's excellent book) and this could be integrated more into the earlier discussion. Kindest regards

Author Response

Dear reviewer, 

thank you for your time in reading our paper and giving detailed feedback. Please find attached our responses to yours as well as the other reviewers. 

With kind regards

Reviewer 3 Report

Comments and Suggestions for Authors

This paper explores the application of an intersectional approach to education and training around violence and abuse, and offers important and insightful key findings and lessons learned from a large multi-site project. The project offers a unique opportunity to learn from the experiences of the multiple partners that can inform both future development of education and training in this space as well as highlight the contributions an intersectional approach can make to how we understand violence and abuse. 

The article has the potential to contribute more significantly to theoretical discourse on intersectionality but does not explicitly engage with the theoretical literature. There is no mention of the definition or understanding of intersectionality used, any background discussion of the broader theoretical discourse, or any sign posting early on to the reader that the focus is only on LGBTQIA+ intersectionality and not other forms of difference or marginalisation. This lack of positionality limits the article’s contribution to our understanding of applying an intersectional approach to education and training, and addressing this would significantly enhance the contribution of the article.

In the later section of the paper on making practical use of theory, there is a missed opportunity to discuss how the lessons learned for this project may be applied to other forms of difference and marginalisation that intersectionality theory encompasses. Engaging with the literature on areas such as disability, race and culture, or Indigeneity for instance would further enhance the contributions of this paper by linking to broader understandings of intersectionality and why this approach is both necessary and impactful in improving our understandings of and responses to violence and abuse. The conclusion raises some of these issues around our internalised biases and the hegemonic power structures we live and work in, and this needs to be present throughout the paper.

In the introduction, the authors indicate that it has been 7 years since the project was completed but that the lessons are still relevant, but do not go on to say why. It would help the reader to understand whether there has been much other published work in this area in the meantime (ie whether this paper is filling a large gap) or whether contextual factors have remained largely unchanged and therefore the lessons still apply in that regard. There are also a couple of areas early on in the paper where clarity is required, eg at line 27 the partners are introduced but it is not immediately apparent how those partners are (are the individuals? Organisations? Consortia?), and at line 69 the main activities took place in only 4 of the 6 countries but it is not clear why (or who the other two countries are and why there was less activity).

In section 3 the issues of translation, the section on the experience in Ireland is significantly shorter than the other countries discussed and feels overly simplistic and dismissive – a more fulsome discussion of the intricacies around the language chosen would be helpful, particularly as they are the only site that used GBV rather than GRV.

In section 4 in the results and findings there are some claims at lines 340-343 abut social media, but it is unclear if this is a finding from the project or general reflections. If it is the latter some references for these claims are required. This goes for any claims that are made that are not grounded in the project being reported, evidence for the claims must be cited. In the discussion of the Italian experience at lines 386-393 about legal aspects, it is not clear why or how the law ‘could not provide’ the certainty looked for, or indeed what that ‘certainty’ was. Greater explication of these concepts would be helpful.

There are a significant number of minor grammatical and syntax errors throughout the paper (too many to list here), and it requires a close edit to identify and rectify these.

Overall this paper offers some important insights and learnings for the future application of intersectional approaches to the design and implementation of education and training about violence and abuse. It does lean heavily on the project report, but incorporating a greater engagement with intersectionality theoretical discourse throughout the paper will ensure it is an original and important contribution to the area.

Author Response

(The authors gave the same response as above.)

Round 2

Reviewer 1 Report

Comments and Suggestions for Authors

It can be published.